# Calcium Pyrophosphate Dihydrate Crystals Increase the Granulocyte/Monocyte Progenitor (GMP) and Enhance Granulocyte and Monocyte Differentiation In Vivo

**DOI:** 10.3390/ijms22010262

**Published:** 2020-12-29

**Authors:** Nobuyuki Onai, Chie Ogasawara

**Affiliations:** Department of Immunology, Kanazawa Medical University, Ishikawa 920-0293, Japan; c-ogasa@kanazawa-med.ac.jp

**Keywords:** calcium pyrophosphate dihydrate crystals, granulocyte, monocyte

## Abstract

Calcium pyrophosphate dihydrate (CPPD) crystals are formed locally within the joints, leading to pseudogout. Although the mobilization of local granulocytes can be observed in joints where pseudogout has manifested, the mechanism of this activity remains poorly understood. In this study, CPPD crystals were administered to mice, and the dynamics of splenic and peripheral blood myeloid cells were analyzed. As a result, levels of both granulocytes and monocytes were found to increase following CPPD crystal administration in a concentration-dependent manner, with a concomitant decrease in lymphocytes in the peripheral blood. In contrast, the levels of other cells, such as dendritic cell subsets, T-cells, and B-cells, remained unchanged in the spleen, following CPPD crystal administration. Furthermore, an increase in granulocytes/monocyte progenitors (GMPs) and a decrease in megakaryocyte/erythrocyte progenitors (MEPs) were also observed in the bone marrow. In addition, CPPD administration induced production of IL-1β, which acts on hematopoietic stem cells and hematopoietic progenitors and promotes myeloid cell differentiation and expansion. These results suggest that CPPD crystals act as a “danger signal” to induce IL-1β production, resulting in changes in course of hematopoietic progenitor cell differentiation and in increased granulocyte/monocyte levels, and contributing to the development of gout.

## 1. Introduction

Traditionally, inflammation involves a biological reaction involving the activation of immune cells, such as macrophages, neutrophils, and other cells, in response to the presence of pathogenic substances originating from microorganisms or by cells that have been affected by pathogenic microorganisms that have invaded the body, such as bacteria and viruses [1]. These microorganism infections trigger innate immunity to eliminating the microorganisms. In the innate immunity, the immune cells recognize pathogenic substances through sensors and then engage in phagocytosis or produce inflammatory cytokines to activate lymphocytes and adaptive immunity and eliminate the pathogenic microorganisms [2]. However, sterile inflammation in the absence of pathogenic microorganisms has garnered research attention in recent years [3,4]. Immune cells also recognize endogenous host-derived damage-associated molecular patterns (DAMPs) and produce inflammatory cytokines, chemokines, and mediators [5]. This sterile inflammation has been shown to play an important role in the onset of pathological states in various diseases such as cancer, lifestyle-related disorders, and cardiovascular diseases [6,7]. One molecular mechanism responsible for sterile inflammation is the inflammasome, an inflammation-inducing pathway associated with innate immunity, which has been of growing interest [6,7].

Pseudogout is caused by calcium pyrophosphate dihydrate (CPPD) crystals deposition in the joint [8]. The CPPD crystals induce production of inflammatory cytokines, chemical mediators, and neutrophil extracellular traps and activation of chondrocytes in the joint, thereby progressing the pseudogout [8,9,10]. These crystal products act as danger signals and activate the NLR family, pyrin domain containing 3 (NLRP3)-inflammasome. This inflammasome goes on to activate the downstream adapter molecule Apoptosis-associated speck-like protein containing a CARD (ASC )and the cystine protease caspase-1, and induces inflammation via the release of IL-1β and IL-18 [11,12,13]. Although it is known that accumulation of CPPD crystals mobilizes granulocytes into the joint capsule of the pseudogout, the mechanism and effects of this on hematopoietic cells have not yet been characterized [14]. Thus, in this study, we focused on the effect of CPPD crystals on hematopoiesis and analyzed the dynamics of granulocytes, monocytes, and bone marrow progenitor cells in mice administered calcium pyrophosphate crystals (Appendix A).

## 2. Results

### 2.1. Calcium Pyrophosphate Dihydrate (CPPD) Crystal Administration Temporarily Reduces White Blood Cell Count

To investigate the effect of CPPD crystals in vivo, peripheral blood was collected over time following the intra-bone-marrow injection of CPPD in wild-type mice, and the white blood cell (WBC), red blood cell (RBC), and platelet counts were measured (Figure 1A). The WBC count was decreased by approximately 30% 1–3 days after CPPD administration, and it subsequently recovered beginning on day 5, ultimately recovering to nearly the same level as the control on day 7 (Figure 1B). In contrast, there were no changes in the RBC and platelet counts to 7 days after calcium pyrophosphate crystals were administered (Figure 1C,D).

### 2.2. CPPD Crystal Administration Increases Granulocytes and Monocytes

Next, we addressed the effect of CPPD administration on the myeloid cell expansion and performed FCM analysis. Using a flow cytometer, a significant increase in Ly6G^+^CD11b^+^ granulocytes and Ly6G^int^CD11b^+^ monocytes in the spleen and peripheral blood was observed after CPPD was administered (Figure 2A,B). Granulocytes increased approximately 10-fold and approximately 3-fold in the spleen and peripheral blood, respectively, when compared with their levels in control mice on post-administration day 3 (Figure 2A,C). Similarly, monocytes increased approximately 1.5-fold in both the spleen and peripheral blood compared to their levels in control mice (Figure 2B–D). The observed increases in granulocytes and monocytes were dependent on the concentration of CPPD (Figure 2C). Increases in granulocytes and monocyte counts in the peripheral blood peaked three days after administration of CPPD and returned to baseline levels by post-administration day 7 (Figure 2D). In addition to myeloid cell expansion, B-cell and T-cell count was decreased by approximately 35–50% in the peripheral blood after administration of CPPD (Figure 2E). In contrast, there was no change in conventional dendritic cell (cDC), plasmacytoid dendritic cell (pDC), B-cell, and T-cell levels in the spleen (Figure 2F). These results confirmed that CPPD administration specifically increased granulocyte and monocyte counts.

### 2.3. CPPD Administration Increases Granulocyte/monocyte Progenitor Cell Populations in the Bone Marrow

CPPD administration was found to specifically increase granulocyte and monocyte counts; therefore, the numbers of hematopoietic stem cells and progenitor cells in the bone marrow, from which granulocytes and monocytes differentiate, were investigated.

The lineage^−^Sca-1^+^c-kit^+^ (LSK) fraction was enriched with hematopoietic stem cells and hematopoietic progenitor cells in the bone marrow [15,16], LSK fraction and the populations of Lin^−^Sca-1^−^c-kit^+^CD34^+^FcγR^+^ granulocytes/monocyte progenitors (GMPs) were significantly increased (Figure 3A–C) (17). Meanwhile, Lin^−^Sca-1^−^c-kit^+^CD34^-^FcγR^−^megakaryocyte/erythrocyte progenitors (MEPs) decreased, and the number of Lin^−^Sca-1^−^c-kit^+^CD34^+^FcγR^−^common myeloid progenitors (CMPs) remained unchanged (Figure 3A–C). Increases in GMP and decreases in MEP counts in the BM peaked three days after CPPD administration and returned to baseline levels by post-administration day 5 (Figure 3D). These result confirmed that CPPD administration specifically increased GMP and deceased MEP numbers.

### 2.4. CPPD Administration Induces IL-1β Production in the Bone Marrow

CPPD activate the NLRP3-inflammasome and induces the IL-1β production in the human monocyte cell line [12]. Therefore, serum IL-1β level was investigated. The concentration of IL-1β was significantly increased depending on the concentration of CPPD compared with that in the control mice on post-administration day 1 (Figure 4A), and it returned to baseline levels by post-administration day 5 (Figure 4B).

## 3. Discussion

In this study, administration of CPPPD crystals in mice, the causative metabolite of pseudogout [8], was found to increase granulocyte and monocyte counts in the peripheral blood and spleen in a concentration-dependent manner. Concomitantly, administration of CPPD induced significant reduction of CD19^+^ B cells and CD3^+^ T cells in the PB. These lymphocytes are a major fraction of WBC; therefore, WBC count was reduced after CPPD administration. Additionally, GMP counts, from which these myeloid cells differentiate [17], significantly increased, whereas MEP counts, which are the origin of erythroid cells [17], decreased in the BM after CPPD administration. Furthermore, the fact that no changes in the numbers of dendritic cell subsets and lymphocyte in the spleen were observed following CPPD administration suggested that CPPD administration specifically enhances GMPs and induces granulocyte and monocyte differentiation. In addition, the serum level of IL-1β was significantly increased in a concentration-dependent manner and it returned to baseline levels by post-administration day 5.

CPPD is recognized as a danger signal by macrophages, and it activates the NLRP3-inflammasome to produce bioactive IL-1β. These were shown in ex vivo experiments using human primary monocytes and monocyte cell lines [11,12,13]. Chronic injection of IL-1β induces the expansion of myeloid-biased hematopoietic progenitors and myeloid cells, and reduces the number of lymphoid cells in the mice [18,19,20]. IL-1β inhibits the B cell differentiation ex vivo in bone-marrow culture [21].

In this study, we found that IL-1β production and expansion of GMPs and granulocytes and monocytes increased in vivo after CPPD administration. It will be useful to determine whether CPPD-induced IL-1β directly acts on CMP and GMP in vivo using IL-1R conditional knockout mice. However, there is no available mouse system to knock down the IL-1R gene on the CMP and GMP. In addition, it might be helpful to address the correlation between IL-1 level and the severity of pseudogout in the patients for prophylactic and therapeutic clinical settings.

Based on the results, we propose the following scenario: In the first step, the CPPD crystal forms within the joint. CPPD is recognized by macrophages in the bone marrow, and it induces IL-1β production. CPPD-induced IL-1β acts on hematopoietic stem cells and GMPs to induce granulocyte and monocyte differentiation, suggesting a model in which the same cell group is mobilized locally in joints. In this model, sterile inflammation is induced by other danger signals, and because of inflammatory cytokine production and enhancement of myeloid cell differentiation, this activity may contribute to the onset of pathological states in pseudogout [22,23].

## 4. Material and Method

### 4.1. Mice

C57BL/6 (B6) mice were purchased from Japan SLC Inc. All mice were maintained in our specific pathogen-free facility, and all experiments using mice were approved by the Animal Care Committee of Kanazawa Medical University (2017-29 on 18 April 2018).

### 4.2. Reagent

CPPD was purchased from Invivogen (# tlrl-cppd, Invivogen, San Diego, CA, USA) and resuspended in PBS (−). The CPPD was injected into the bone marrow cavity of the mice.

### 4.3. Peripheral Blood Cell Count

White blood cells (WBCs), red blood cells (RBCs), and platelet counts were determined using the pocH-100iV auto cell counter (Sysmex, Kobe, Japan).

### 4.4. FCM Analysis

Spleens and lymph nodes were cut in small fragments and digested under repeated agitation for 30 min at 37 °C in RPMI 1640 medium supplemented with 10% FCS, 1 mg/mL collagenase (collagenase D; Roche Diagnostic Systems, Basel, Switzerland), and 10 μg/mL DNase (DNase I from bovine pancreas grade II; Roche Diagnostic Systems). Debris was removed by filtration, and red cells were lysed osmotically. Nucleated cells were stained as indicated with fluorochrome-conjugated Abs and were analyzed by FACS. Immediately after euthanasia, bone marrow (BM) cells were harvested from the mouse femurs, tibias, spine, and iliac crest, and were then filtered and washed in cold PBS containing 2 mM EDTA. BM cells were labeled with PE-Cy5-conjugated antibodies against lineage antigens consisting of CD3ε (#100310, clone 145-2C11), CD4 (#100410, clone RM4.5), CD8α (#100710 clone 53-6.7), B220 (#103210 clone RA3-6B2), MHC class II (#107612 clone M5/114.15.2), CD11b (#101210 clone M1/70), Gr-1 (#108410 clone RB6-8C5), TER119 (#116210 clone TER119), NK1.1 (#108716 clone PK136). To detect progenitors, the BM lineage-negative (Lin^−^ cells were stained with FITC-anti-CD34 (#11-341-82 clone RAM34; eBioscience), PE/Cyanine7-anti-Sca-1 (#108114 clone D7), APC-anti-c-Kit (#106812 clone 2B8), APC/Cyanine7- FcγRⅡ/Ⅲ (#101328 clone 93). LSK cells were defined as Lin^−^Sca-1^+^c-Kit^+^ cells, and myeloid progenitors were defined as CMPs (Lin^−^c-kit^+^Sca-1^−^CD34^+^FcγR^−^), GMPs (Lin^−^c-kit^+^Sca-1^−^CD34^+^FcγR^+^), and MEPs (Lin^−^c-kit^+^Sca-1^−^CD34^−^FcγR^−^). Antibodies against the following molecules were used for flow cytometric analysis: PE-Ly6G (#127607 clone 1A8), PE-Cyanine7-CD11b (#101216 clone M1/70), APC-CD11c (#117310 clone N418), PB-CD317 (#127108 clone 129C1), APC/Cyanine7-TER119 (#116223 clone TER119), APC-CD3ε (#100312 clone 145-2C11), FITC-CD4 (#116004 clone RM4.5), PE-CD8α (#100708 clone 53-6.7), PE/Cyanine7-CD19 (#115520 clone MB19-1), APC/Cyanine7-NK1.1 (#108024 clone PK136). All antibodies were purchased from Biolegend (San Diego, CA, USA). The stained cells were analyzed on a FACSCanto II (BD) in conjunction with FlowJo software (TreeStar).

### 4.5. ELISA

IL-1β serum levels were determined by ELISA kits according to the manufacturer’s instructions (Biolegend, San Diego, CA, USA).

### 4.6. Statistical Analysis

We evaluated the statistical significance of the differences between obtained values using the two-tailed Student’s *t-*test. A *p-*value < 0.05 was considered significant.

## 5. Conclusions

The in vivo administration of CPPD induced expansion of granulocyte/monocyte progenitors (GMPs) in the BM, and their progenies, granulocytes and monocytes in the spleen and peripheral blood, production of IL-1β, and reduction of MEPs. It has been reported that CPPD acts as a danger signal, activates the NLRP3-inflammasome, and induces IL-1 production. In addition, IL-1 acts on hematopoietic stem cells to induce myeloid cell differentiation. Based on these results, CPPD crystals act as a “danger signal” to induce IL-1β production, resulting in expansion of GMPs and in increased granulocyte/monocyte levels, and contributing to the development of gout.

## Figures and Tables

**Figure 1 ijms-22-00262-f001:**
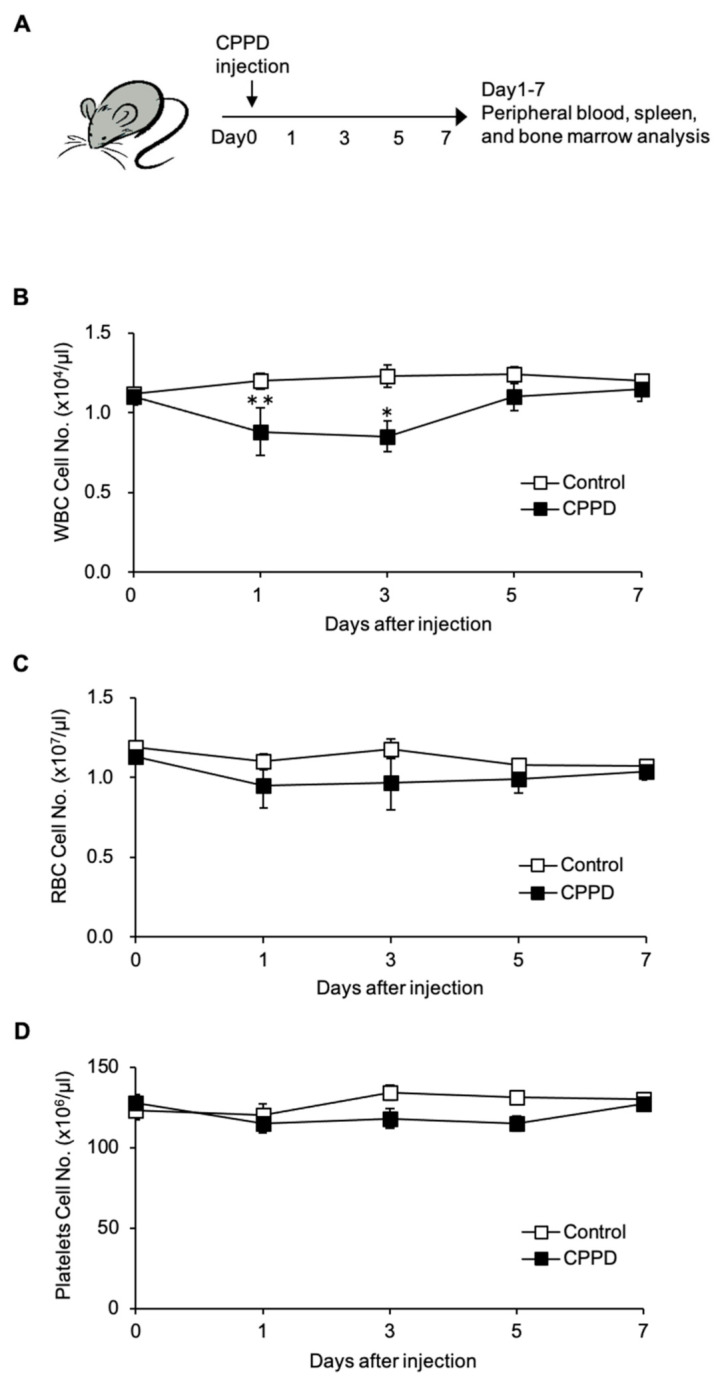
Calcium pyrophosphate dihydrate (CPPD) injection induces reduction of white blood cells (WBCs) in the peripheral blood. (**A**) Experimental design for in vivo injection of CPPD. (**B**–**D**) The time course of cell number of WBCs (**B**), red blood cells (RBCs) (**C**), and platelets in the peripheral blood (PB) (**D**) were analyzed after vehicle (PBS) or CPPD (100 μg) injection. Data represent the mean ± s.d. of three independent experiments. * *p* < 0.01, ** *p* < 0.05.

**Figure 2 ijms-22-00262-f002:**
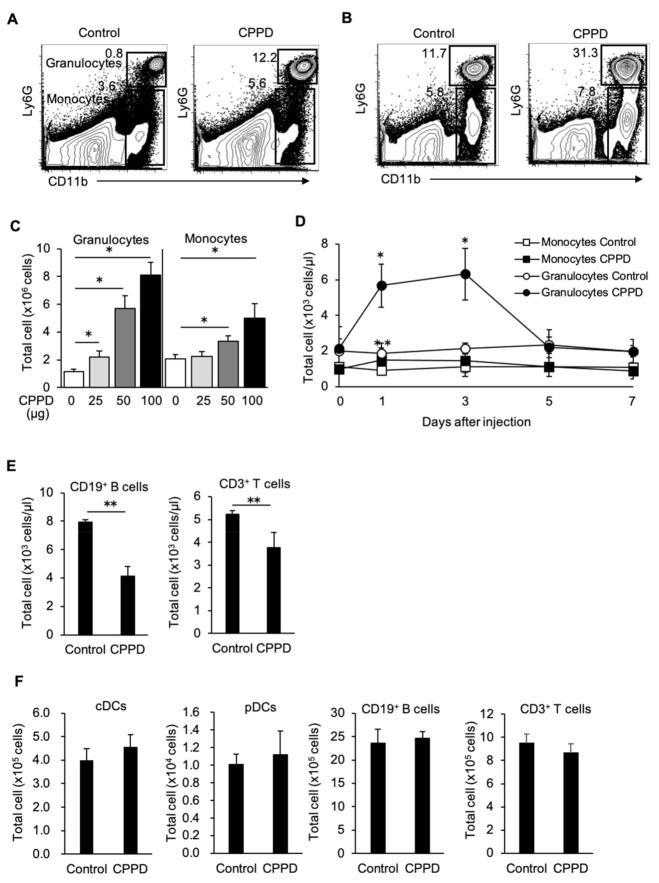
CPPD injection induces myeloid cell expansion in vivo. (**A,B**) The percentage of granulocytes (Ly6G^+^CD11b^+^) and monocytes (Ly6G^int^CD11b^+^) in the spleen (**A**) and peripheral blood (**B**) of the mice 3 days after the CPPD injection. (**C**,**D**) The total cell number of granulocyte and monocytes in the spleen (**C**) and PB (**D**) after CPPD injection of the indicated dose and at the indicated time course. (**E**,**F**) Total cell number of CD19^+^ B cells, and CD3^+^ T cell in the peripheral blood **(E**) and conventional dendritic cells (cDCs), plasmacytoid DC (pDC), and CD19^+^ B cells, and CD3^+^ T cells in the spleen (**F**). Data represent the mean ± s.d. of three independent experiments. * *p* < 0.01, ** *p* < 0.05.

**Figure 3 ijms-22-00262-f003:**
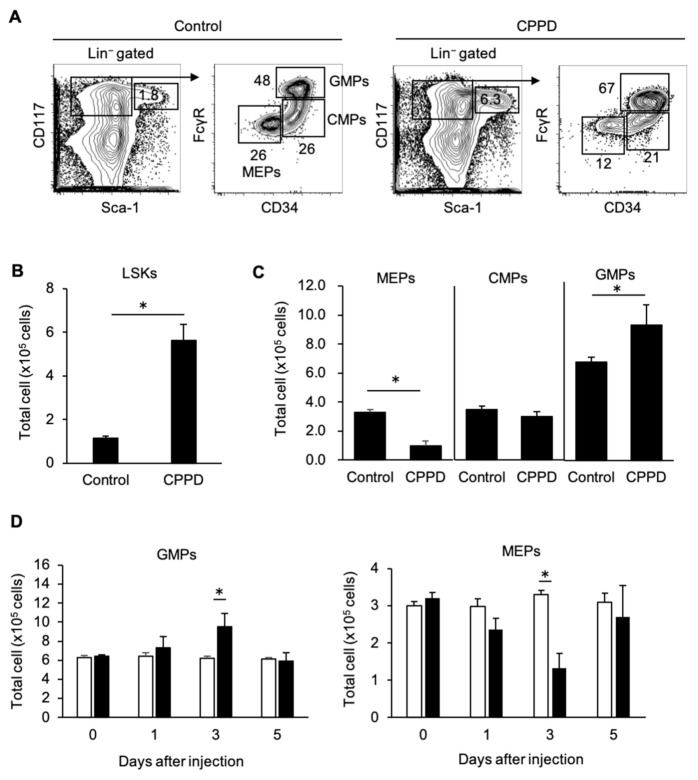
CPPD injection induces granulocytes/monocyte progenitors (GMPs) expansion and megakaryocyte/erythrocyte progenitors (MEPs) reduction in vivo. (**A**–**D**) The percentage (**A**) and total cell number of lineage^−^ Sca-1^+^c-kit^+^ (LSK) cells (**B**) and megakaryocyte/erythrocyte progenitors (MEPs), common myeloid progenitors (CMPs), and granulocytes/monocyte progenitors (GMPs) in the bone marrow 3 days after vehicle (PBS) or CPPD (100μg) injection (**C**). (**D**) Time course of absolute cell number of GMP and MEP in the bone marrow after CPPD administration. Data represent the mean ± s.d. of three independent experiments. * *p* < 0.01.

**Figure 4 ijms-22-00262-f004:**
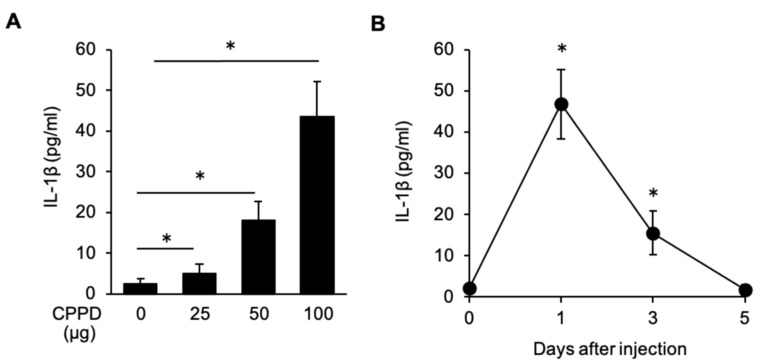
CPPD injection induces IL-1β production in vivo. (**A**,**B**) Serum IL-1β production in WT mice 1 day after CPPD administration at the indicated concentration. (**B**) Kinetics of serum IL-1β production after CPPD administration. Data represent the mean ± s.d. of three independent experiments. * *p* < 0.01.

## Data Availability

Not applicable.

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
