# Peer review of "Calcium Pyrophosphate Dihydrate Crystals Increase the Granulocyte/Monocyte Progenitor (GMP) and Enhance Granulocyte and Monocyte Differentiation In Vivo"

_ijms, 2020, doi:10.3390/ijms22010262_

Round 1

Reviewer 1 Report

In this study, the authors demonstrate CPPD crystals increase the granulocyte/monocyte progenitor and enhance their differentiation. The aim is clear, and findings are interesting. However, the detail raised some questions and recommendations.

  1. Please clarify the route of administration of CPPD crystals on mice in methodology (intra-peritoneal, intra-articular, or intra-muscular injection?).
  2. The clone and catalog number of antibodies are needed.
  3. Are the results statistically significant in figure 1A and 2D? If they are, please mark the statistical significance.
  4. The results showed the increase of granulocytes in peripheral blood (figure 2D) and granulocytes/monocytes progenitors in bone marrow (figure 3C) on the 3 days after CPPD injection. However, the result 2.1 revealed that the peripheral WBC count was decreased by approximately 30% on 1-3 days after administration (figure 1A). Please show the differential count of WBC on 1-3 days after administration of CPPD and discuss the relationship.

Author Response

Dear Reviewer 1

Thank you for your review and comments on our manuscript (ijms-1038738),

 “Calcium pyrophosphate dihydrate crystals increase the granulocyte/monocyte progenitor (GMP) and enhance granulocyte and monocyte differentiation in vivo.”

We would like to express our sincere thanks for the opportunity to revise our manuscript. According to the Reviewers’ comments, we carefully revised our manuscript in a point-by-point fashion.

Comments and Suggestions for Authors

In this study, the authors demonstrate CPPD crystals increase the granulocyte/monocyte progenitor and enhance their differentiation. The aim is clear, and findings are interesting. However, the detail raised some questions and recommendations.

  1. Please clarify the route of administration of CPPD crystals on mice in methodology (intra-peritoneal, intra-articular, or intra-muscular injection?).

A: In the page 10 line 8-9, we described the route of administration of CPPD crystal. The CPPD crystal were inject into the bone marrow cavity of the mice.

  1. The clone and catalog number of antibodies are needed.

A: In the page 11 line 8 to page 12 line 5, we added the clone and catalog number of antibodies.

  1. Are the results statistically significant in figure 1A and 2D? If they are, please mark the statistical significance.

A: We performed statistical analysis, then we marked the statistical significance in the Fig. 1A and 2D.

  1. The results showed the increase of granulocytes in peripheral blood (figure 2D) and granulocytes/monocytes progenitors in bone marrow (figure 3C) on the 3 days after CPPD injection. However, the result 2.1 revealed that the peripheral WBC count was decreased by approximately 30% on 1-3 days after administration (figure 1A). Please show the differential count of WBC on 1-3 days after administration of CPPD and discuss the relationship.

A: In the page 6 lines 7-9. As we agreed to the referee’s comment, we showed the number of B cells and T cells in peripheral blood (Figure 1 E). The number of these lymphocytes, which are a major fraction in the PB, were significantly reduced after CPPD administration, therefore, total WBC counts was reduced. We also added discussion (page 8 lines 4-7.).

We look forward to hearing from you.

Sincerely,

Nobuyuki Onai, Ph.D.

Reviewer 2 Report

Abstract: “as a result of metabolic disorders affecting the joints” is too wordy.

Introduction: Wording, syntax, and phraseology could be more precise and specific in fewer words. Pseudogout can result from CPPD crystal formation.

Present the purpose of this study and why it is novel from prior studies on the subject.

Comment-Methods should appear in a separate paragraph including the study design, etc.  The results section should contain only results. If desired, the different sections could have methods and results under one heading. It is unusual to present the methods at the end of the paper where conclusions are usually found. Please explain or change.

  1. In response to CPPD administration, WBC count was decreased by approximately 30% 1–3 days after

administration, and subsequently recovered

  1. After CPPD, granulocytes and monocytes increased in the peripheral blood and spleen compared to control mice.
  2. This was also true for GMPs, while megakaryocyte/RBC progenitor cells decreased.
  3. CPPD injection induced IL-B production in vivo.

Limitations: A limitations and delimitations section is needed. It should specifically mention the results require more work and should not be generalized beyond the species used herein.

Discussion:

Suggest a cartoon or central graphic illustration of the theoretical concept of the study with addenda attached for pertinent comments and observations.

There are only suggestions to improve the manuscript. If completed they should raise the ratings given 

Suggest additional recent citations to the literature for the discussion of inflammation. The references used are general discussions. Although they are welcome, more specific references relevant to the sentence they are used to support should also appear.

A beginning two sentences should define innate and adaptive immunity and their significance in this study.

Another paragraph should compare (in detail) these results versus other study results on the same topic.

The conclusions usually follow at the end of the paper.

Author Response

Dear Reviewer2

Thank you for your review and comments on our manuscript (ijms-1038738),

 “Calcium pyrophosphate dihydrate crystals increase the granulocyte/monocyte progenitor (GMP) and enhance granulocyte and monocyte differentiation in vivo.”

We would like to express our sincere thanks for the opportunity to revise our manuscript. According to the Reviewers 2’ comments, we carefully revised our manuscript in a point-by-point fashion.

Comments and Suggestions for Authors

Abstract: “as a result of metabolic disorders affecting the joints” is too wordy.

A: In the page 2 lines 1-2. We agree reviewers’ comments. We have changed the wording to, “Calcium pyrophosphate dihydrate (CPPD) crystals are formed deposited locally within the joints, leading to pseudogout.’

Introduction: Wording, syntax, and phraseology could be more precise and specific in fewer words. Pseudogout can result from CPPD crystal formation.

A: In the page 4 lines 1-10. We agree reviewers’ comments. We have changed the introduction.

Present the purpose of this study and why it is novel from prior studies on the subject.

A: In the page 4 lines 10-13. We agree reviewers’ comments. We added the purpose and novelty of this study and also added the supplementary figure to explain for the purpose of the study.

Comment-Methods should appear in a separate paragraph including the study design, etc. 

A: In the page 5 lines 4-7. We agree reviewers’ comments. We added the experimental design for the study and Figure 1A.

The results section should contain only results. If desired, the different sections could have methods and results under one heading. It is unusual to present the methods at the end of the paper where conclusions are usually found.

Please explain or change.

  1. In response to CPPD administration, WBC count was decreased by approximately 30% 1–3 days after administration, and subsequently recovered
  1. After CPPD, granulocytes and monocytes increased in the peripheral blood and spleen compared to control mice.
  2. This was also true for GMPs, while megakaryocyte/RBC progenitor cells decreased.
  3. CPPD injection induced IL-B production in vivo.

A: In the page 8 lines-2-15 and page 9 lines 11-18. We agree reviewers’ comments. We added the discussion and our explanation of the changing the haematopoiesis after CPPD administration. We added the only results in the results section.

Limitations: A limitations and delimitations section is needed. It should specifically mention the results require more work and should not be generalized beyond the species used herein.

A: In the page 9 lines 5-10. We agree reviewers’ comments. We added and discussed the limitation and delimitations of our study.

Discussion:

Suggest a cartoon or central graphic illustration of the theoretical concept of the study with addenda attached for pertinent comments and observations.

A: In the page 4 lines 8-13. We agree reviewers’ comments. We added the theoretical concept of the study and supplementary figure.

There are only suggestions to improve the manuscript. If completed they should raise the ratings given 

Suggest additional recent citations to the literature for the discussion of inflammation. The references used are general discussions. Although they are welcome, more specific references relevant to the sentence they are used to support should also appear.

A: In the page 3-4. We agree reviewers’ comments. We improved introduction section.

A beginning two sentences should define innate and adaptive immunity and their significance in this study.

A: In the page 4 lines 6-10. We agree reviewers’ comments. We added the definition of innate and adaptive immunity.

Another paragraph should compare (in detail) these results versus other study results on the same topic.

A: In the page 8 line 16 to page 9 line 10. We agree reviewers’ comments. We added the comparison of our study and previous studies.

The conclusions usually follow at the end of the paper.

A: In the page 13. We added the conclusion of our study.

We look forward to hearing from you.

Sincerely,

Nobuyuki Onai, Ph.D.

Round 2

Reviewer 2 Report

This manuscript is now a pleasure to read. It is vastly improved.

Thank you.